# Addition of Chelators Increased the Stability of Black Rice Anthocyanins against the Metallic Ions in Tap Water and Improved the Coloration of Steamed Cold Noodles

**DOI:** 10.3390/foods11213392

**Published:** 2022-10-27

**Authors:** Yi Zheng, Ling Jiang, Chun-Zhi Zhang, Guo-Qing Huang, Li-Ping Guo, Jun-Xia Xiao

**Affiliations:** 1College of Food Science and Engineering, Qingdao Agricultural University, Qingdao 266109, China; 2Special Food Research Institute, Qingdao Agricultural University, Qingdao 266109, China

**Keywords:** black rice anthocyanins, metal chelator, stability, steamed cold noodles, antioxidant activity

## Abstract

The anthocyanins in black rice extract (BRA) are sensitive to metallic ions, which restrict its application in the coloration of steamed cold noodles in China that uses tap water as the solvent. Food-grade chelators were added to check if they could increase the stability of BRA. The results indicated that the color decay of BRA in tap water was mainly caused by Fe^3+^, Cu^2+^, and Fe^2+^, and the addition of chelators could effectively antagonize this effect. Coloration with the BRA solution containing the optimized chelator formulation of 0.01% ethylenediaminetetraacetic acid disodium, 0.08% sodium hexametaphosphate, and 0.064% sodium tartrate conferred comparable appearance and chromatic attributes with those of the noodle colored by deionized water-dissolved BRA. The steamed cold noodles colored by the chelators-containing BRA exhibited increased springiness and decreased starch retrogradation, and possessed potential health functions due to its slightly increased resistant starch content and markedly enhanced antioxidant capacity. Hence, the addition of chelators is a feasible way to increase the color stability of BRA in tap water, and the chelators-supplemented BRA could be used to produce steamed cold noodles with attractive color and health benefits.

## 1. Introduction

Color is an important attribute for the sensory quality of foods. Multiple colorants have been approved to be used in foods, among which synthetic colorants are often selected due to their low cost and high stability. However, natural pigments are attracting more and more interest all over the world because of their health benefits and safety [1,2]. Several categories of natural pigments, including anthocyanins, carotenoids, betalains, and chlorophylls, have been widely used in the food industry [3], among which anthocyanins obtained from flowers, fruits, leaves, and even whole plants are the most extensively studied [4]. Multiple plant extracts have been approved by the regulations of China to be used as colorants in the food industry.

Anthocyanins are extremely unstable against a wide range of parameters such as light, pH, temperature, sugars, vitamin C, oxygen levels, sulfur dioxide or sulfites, enzymes, co-pigments, and metallic ions [5]. Due to the low stability during processing and storage, the direct use of anthocyanins in food formulations, especially in water systems, is challenging [6]. In order to avoid the adverse effects of metallic ions, deionized or ultrapure water is often used when anthocyanins are used as colorants [7,8]. However, tap water is widely utilized in food processing and daily life in China. According to the Chinese National Standard for Drinking Water Quality (GB 5749-2006), trace amounts of metallic ions, including Al^3+^, Fe^2+^, Cu^2+^, Mn^2+^, and Zn^2+^, are allowed in tap water [9]. The presence of these metals could destabilize anthocyanins and limit the application of anthocyanins-rich plant extracts as food colorants. Hence, measures to improve the stability of anthocyanins against the metals are of great importance to extend the application of the extracts in food processing. 

Multiple methods, including co-pigmentation, exclusion of oxygen, and microencapsulation, have been proposed to increase the stability of anthocyanins against pH, light exposure, and temperature [10], but reports concerning improving their stability against metallic ions are nearly unavailable, except two recent works revealing that the formation of nanocomposites with peptides [11] and amylopectin [12] was effective in addressing this issue. Chelators could form stable complexes with metallic ions and thereby suppress their chemical activities. Multiple chelators have been allowed to be used in foods according to the Chinese National Standard for Uses of Food additives (GB 2760-2014) [13]. Hence, the addition of the chelators to anthocyanin-rich extracts might be a feasible and industrially scalable way to increase their stability in tap water. 

Black rice has a high anthocyanins content in the pericarp layer, and the black rice extract (BRA) has been reported able to effectively reduce cholesterol level, suppress tumor growth, delay aging, and prevent cardiovascular diseases [14]; meanwhile, the extract has excellent antioxidant strength that is twice that of blueberries [15]. The application of BRA as a food colorant has been a hot topic in recent years [16]. Its stability against several food processing conditions such as pH and heating has been evaluated [17] and co-pigmentation in combination with microencapsulation has been applied to increase its stability during storage and light exposure. Nevertheless, measures to increase its stability against metallic ions or in tap water have not been investigated yet.

Steamed cold noodles (namely Liangpi in Chinese Pinyin) are a traditional food in many areas of China that are produced by steaming rice or wheat starch followed by cooling to room temperature and cutting into strips. The coloration of steamed cold noodles is an effective method to attract the interests of consumers and increase the diversity of this product. However, steamed cold noodles are often handmade at home or by peddlers without industrialized food processing [18], and tap water is widely used in its production, which restricts the application of BRA in its coloration. 

Hence, the purpose of the current work is to increase the stability of a commercial anthocyanins-rich BRA against the metallic ions in tap water by adding five chelators approved by the Chinese National Standard for Uses of Food additives (GB 2760-2014) [13] and explore the application of the resultant BRA in the coloration of steamed cold noodles. We believe that the work could extend the application of commercial anthocyanins-rich extracts in the coloration of foods in China or other regions that use tap water or hard water as the solvent.

## 2. Materials and Methods

### 2.1. Materials

A commercial black rice extract (BRA) with anthocyanin content greater than 25% was purchased from Haoyukangze Co., Ltd. (Xi’an, China). Wheat starch was purchased from Haining Fengyuan Food Co., Ltd. (Jiaxing, China). Alpha-amylase was purchased from Macklin Biochemical Technology Co., Ltd. (Shanghai, China). Alpha-glucosidase was purchased from BOSF Biotechnology Co., Ltd. (Hefei, China). Other chemicals were of analytical grade.

### 2.2. Color Stability of the BRA Solution against Metallic Ions

According to the Chinese National Standard for Drinking Water Quality (GB 5749-2006) [9], nine kinds of metallic ions, including Na^+^, Fe^3+^, Al^3+^, Cu^2+^, Ca^2+^, Mg^2+^, K^+^, Zn^2+^, and Fe^2+^, in maximum concentrations 200 mg/L, 0.3 mg/L, 0.2 mg/L, 1 mg/L, 180 mg/L, 200 mg/L, 735 mg/L, 1 mg/L, and 1 mg/L respectively, are allowed in the tap water of China. Hence, the effects of these metals on the color stability of BRA were investigated first based on their maximum concentrations. BRA was dissolved in deionized water to yield the 0.02% (*w/v*) solution. Then, the above metals in their chloride forms were dissolved in the BRA solution to generate 5 serial concentrations. The mixtures were left in room temperature and the retention rates of anthocyanins were measured at 0 h, 8 h, 16 h, 24 h, and 32 h. The 0.02% (*w/v*) BRA solution dissolved in deionized water was used as control.

### 2.3. Screening of Chelators

According to the Chinese National Standard for Uses of Food additives (GB 2760-2014) [13], five chelators, including ethylenediaminetetraacetic acid disodium salt (EDTA), sodium hexametaphosphate (SHMP), sodium tartrate (ST), sodium citrate (SC), and sodium gluconate (SG), were selected to investigate their protective effects on the color stability of BRA. That is, one of the five chelators and BRA were dissolved in deionized water until their concentrations reached 0.01% (*w/v*) and 0.02% (*w/v*), respectively. Then, the metals that could cause BRA color decay were added until their concentrations reached the maximum values allowed by GB 5749-2006 [9]. The mixtures were left at room temperature and the retention rates of anthocyanins were measured at 0 h, 8 h, 16 h, 24 h, 32 h, and 40 h.

### 2.4. Optimization of Chelator Combination

Three most effective chelators were screened from the above step, and each one was selected as the major ingredient to optimize their formulations in stabilizing BRA color in tap water. That is, the three chelators of selected concentrations and BRA were co-dissolved in tap water in a fixed BRA concentration 0.02% (*w/v*). The mixtures were left at room temperature and the retention rates of anthocyanins were measured at 0 h, 8 h, 16 h, 24 h, 32 h, and 40 h.

### 2.5. Determination of Anthocyanins Retention Rate

The BRA powder was dissolved in deionized water to yield the 0.02% (*w/v*) solution and then scanned from 400 nm to 700 nm with a UV-2000 spectrophotometer (Unico, Shanghai, China) to determine the maximum absorption wavelength (λ_max_). The absorbances of the sample solutions were read in λ_max_ (A_t_) and compared with that of the solution without any metals (A_0_). Then, the anthocyanins retention rate was calculated using the following equation [15]:$$\mathrm{Anthocyanins}\;\mathrm{retention}\;\mathrm{rate}\;\left(\%\right)=\frac{A_{t}}{A_{0}}\times 100$$

### 2.6. Coloration of Steamed Cold Noodles

BRA as well as the optimized chelators formulations were dissolved in tap water to generate the 0.02% (*w/v*) colorant solution. The BRA solutions in deionized and tap water of the same concentration without any chelators were used as controls. The BRA solutions were blended with wheat starch in mass ratio 3:2 and thoroughly mixed. The resultant pastes were then spread out on a flat dish that was covered with a thin layer of olive oil and steamed for 120 s. Afterwards, the plate was taken out and left to cool to room temperature. Then, the starch gel was brushed with olive oil on its surface to prevent adhesion and then cut into strips to yield steamed cold noodles. The subsequent characterizations of the noodles were completed within 2 h.

### 2.7. Chromatic Aberration Determination of Steamed Cold Noodles

A CR-400 Chroma meter (Konica Minolta, Tokyo, Japan) was used to detect the color of steamed cold noodles based on the CIE L*a*b* color system. The three chromatic parameters were measured at three different locations of each sample at room temperature and mean values were recorded. The color differences (ΔE) compared with the steamed cold noodles colored with deionized water-dissolved BRA were calculated according to the following equation [19]:$$\Delta E=\sqrt{{\left(L^{*}-L_{0}\right)}^{2}+{\left(a^{*}-a_{0}\right)}^{2}{+(b}^{*}-b_{0}{)}^{2}}$$

### 2.8. Textural Profile Analysis (TPA) of Steamed Cold Noodles

The texture attributes of steamed cold noodles were determined using a TA.XT Plus texture analyzer (Stable Micro Systems, Surrey, UK) as previously described with some modifications [20]. The samples were cut into 3 cm × 1.5 cm pieces and placed on the stage of the analyzer and the cylindrical probe P/0.5 was selected for the TPA analysis. The pre-test, test, and post-test speeds were all set to 1 mm/s. The strain was set at 50% with a trigger force of 5 g. Five seconds of resting time was followed between the first and second compression.

### 2.9. In Vitro Digestion of Steamed Cold Noodles

The in vitro digestion behavior of steamed cold noodles was performed using the Englyst’s method [21] with slight modifications. Briefly, 1 g of smashed steamed cold noodles was hydrolyzed in vitro using the pancreatin extract and amyloglucosidase in a 37 °C shaking water bath for 20 min. The concentrations of released glucose were monitored at 20 min and 120 min of the digestion process using a d-glucose assay kit in the glucose oxidase method (Nanjing Jiancheng Bioengineering Institute, Nanjing, China). The starch in steamed cold noodles was classified into three types according to their hydrolysis rates, including the rapidly digestible starch (RDS) that was digested within 20 min, the slowly digestible starch (SDS) that was digested between 20 min and 120 min, and the resistant starch (RS) that was undigested after 120 min, and their contents were calculated according to the following equations: $$\mathrm{RDS}=\frac{G_{20}\times 0.9}{\mathrm{TS}}\;\mathrm{SDS}=\frac{{(G}_{120}-G_{20})\times 0.9}{\mathrm{TS}}\;\mathrm{RS}=\frac{(\mathrm{TS}-\mathrm{RDS}-\mathrm{SDS})}{\mathrm{TS}}$$
where 0.9 is the conversion coefficient from starch to glucose, G_20_ and G_120_ are the glucose contents released at 20 and 120 min respectively, and TS represents the total glucose content in the starting steamed cold noodles. 

### 2.10. Determination of Starch Retrogradation in Steamed Cold Noodles

The degree of starch retrogradation in steamed cold noodles was determined by X-ray diffraction (XRD). The fresh steamed cold noodles were stored in a 4 °C refrigerator for 72 h for aging. Then, the samples were freeze-dried, ground, and passed through a 100-mesh sieve. The fine powder was collected, loaded to a D8 Advance Powder X-ray Diffractometer (Brucker, Karlsruhe, Germany), and scanned from 4° to 40° with a step size of 0.02°. The divergence, receiving, and scatter slit widths were set to 1°, 0.02 mm, and 1°, respectively, at 40 kV and 40 mA [22]. The Jade software V6.5 (Jade, Christchurch, New Zealand) was applied to analyze the crystallinity.

### 2.11. Determination of Total Polyphenols in Steamed Cold Noodles

The phenolic compounds in steamed cold noodles were determined following the method described by Gogoi, et al. [23] with some modifications. That is, 5 g of each of the crushed steamed cold noodles were immersed in 50 mL of 95% ethanol and sonicated at 40 KHz and 200 W for 20 min. Then, the suspensions were centrifuged in 5000× *g* for 15 min, and the supernatants were collected for total polyphenols content determination in the Folin–Ciocalteu method. The results were expressed as gallic acid equivalents (GAE). 

### 2.12. Determination of Total Anthocyanins Contents in Steamed Cold Noodles

The contents of total anthocyanins in steamed cold noodles were determined using the pH differential method [24]. That is, each of the steamed cold noodles extracts mentioned above was diluted with both the 0.025 mol/L potassium chloride buffer of pH 1.0 and the 0.4 mol/L sodium acetate buffer of pH 4.5 until their absorbances at 520 nm were within the linear range. Then, the absorbances of two dilutions were read at both 520 and 700 nm against deionized water, and the total anthocyanins content (TA) was expressed as mg of cyanidin-3-glucoside equivalents per kilogram of steamed cold noodles according to the following equation: $$\mathrm{TA}\;(\mathrm{mg}/\mathrm{kg})=\frac{A\times \mathrm{MW}\times \mathrm{DF}\times 1000}{\varepsilon \times C}$$
where A is the difference of the (A_520_–A_700_) value of the extract in pH 1.0 versus the (A_520_–A_700_) value in pH 4.5, MW is the molecular weight of cyanidin-3-glucoside (449.2 g/mol), DF is the dilution factor, ε is the molar extinction coefficient of cyanidin-3-glucoside (26,900 mol^−1^ cm^−1^), and C is the concentration of the steamed cold noodles extract.

### 2.13. Determination of Antioxidant Capacity of Steamed Cold Noodles

The antioxidant capacities of steamed cold noodles were measured according to the method described in a previous study [25]. Briefly, 2.0 mL of the steamed cold noodles extract mentioned above was mixed with the same volume of DPPH methanol solution and incubated in the dark and at room temperature for 30 min. Then, its absorbance (A_sample_) was read at 517 nm against methanol (A_control_), and the DPPH radical scavenging capacity was calculated using the following equation:$$\mathrm{DPPH}\;\mathrm{radical}\;\mathrm{scavenging}\;\mathrm{activity}\;\left(\%\right)=\frac{A_{\mathrm{control}}-A_{\mathrm{sample}}}{A_{\mathrm{sample}}}\times 100$$

### 2.14. Statistical Analysis

Results were reported as mean ± standard deviation of at least three independent experiments. One-way analysis of variance (ANOVA) followed by Tukey’s test (*p* < 0.05) was performed using SPSS V19.0 (SPSS Inc., Chicago, IL, USA).

## 3. Results and Discussion

### 3.1. Effect of Metallic Ions on the Color Stability of BRA

Anthocyanins could bind metallic ions via the catechol B-ring in neutral or mildly acidic solutions [26]. As shown in Figure 1a, the BRA solution exhibited an attractive red color in deionized water; however, the color turned black in tap water or deionized water containing the nine metals allowed by GB 5749-2006 [9], indicating that the presence of metallic ions could cause color decay of the BRA solution, which could limit the application of BRA as a food colorant. The fading of BRA in tap water was greater than that in the metals-containing deionized water, implying that other components in tap water could also cause color decay to BRA.

In preliminary trials, we found that Na^+^, Al^3+^, Ca^2+^, Mg^2+^, K^+^, and Zn^2+^ in their maximum concentrations allowed by GB 5749-2006 [9] had no obvious effects on the appearance or color intensity of the 0.02% (*w/v*) BRA solution. Hence, only the effects of three metallic ions, namely Fe^3+^, Cu^2+^, and Fe^2+^, on the color stability of the BRA solution was investigated as a function of their concentrations. As can be seen in Figure 1b–d, the color intensity of the control BRA solution (dissolved in deionized water) decayed with storage proceeding due to the exposure to light, pH, and temperature and the addition of one of the three metals further accelerated this process in a concentration-dependent manner. These results were consistent with previous reports that the presence of iron caused undesirable blackening of tannin-containing products [27,28] and another report that the addition of Cu^2+^ caused color decay of the cyanidin 3-O-glucoside solution [11]. 

Figure 1b–d also revealed that BRA was the most sensitive to the presence of Cu^2+^, followed by Fe^3+^ and Fe^2+^ in sequence. After incubation with 0.06–0.30 mg/L Fe^3+^, 0.2–1.0 mg/L Cu^2+^, and 0.2–1.0 mg/L Fe^2+^ for 32 h, the anthocyanins retention rates of the three BRA solutions were 83.69–78.17%, 83.16–73.75%, and 80.73–80.08% respectively, which were lower than 84.46% of the control. In addition, by checking the insets in Figure 1, it could be seen that the appearance of the BRA solution in tap water or deionized water containing the mixture of the nine metals was darker than that of the BRA solutions containing one of the three metals, implying that the synergistic or additive fading effect on BRA color possibly exists between the metals, which needs confirmation by extra experiments. 

### 3.2. Effects of Chelators on the Color Stability of BRA against Fe^3+^, Cu^2+^, and Fe^2+^

In this work, a simple and industrially scalable strategy was adopted to increase the color stability of BRA against the metallic ions in tap water. Ethylenediaminetetraacetic acid disodium salt (EDTA), sodium hexametaphosphate (SHMP), sodium tartrate (ST), sodium citrate (SC), and sodium gluconate (SG) could chelate metals and are allowed as food additives according to the regulations in China [13]. Their effects on the color stability of BRA in the presence of Fe^3+^, Cu^2+^, and Fe^2+^ in their maximum allowed concentrations, namely 0.3 mg/L, 1.0 mg/L, and 1.0 mg/L, respectively, were investigated.

As can be seen from Figure 2, the addition of all the chelators effectively improved the color stability of the BRA solution against the three metals, and the effect was closely dependent on the metal and chelator type. Regarding the presence of Fe^3+^, the addition of EDTA, SHMP, ST, SC, and SG conferred the anthocyanins retention rates of 97.66%, 97.51%, 96.34%, 93.3%, and 86.52%, respectively, after storage for 40 h, which were significantly higher than 76.87% of the control. A similar trend was observed in the protection against Cu^2+^, that is, the addition of the five chelators provided the anthocyanins retention rates of 96.99%, 99.33%, 98.56%, 96.09%, and 91.39%, respectively, after storage for 40 h. Nevertheless, a different protection effect was observed in the protection against Fe^2+^. That is, EDTA, ST, SHMP, and SG conferred comparable anthocyanins retention rates of 95.20%, 94.46%, 93.87%, and 93.05%, respectively, while SC was the least effective with an anthocyanins retention of 89.69%. 

In a previous work, it was found that the addition of EDTA showed no effect on the color of strawberry juice during storage at −20 °C and +20 °C [29], which implied that EDTA and other chelators were possibly unable to interact with anthocyanins. Hence, the chelators in this work might exert their effects by chelating with the metallic ions and the addition of chelators might be a feasible way to increase the stability of anthocyanins in tap water. Because EDTA, SHMP, and ST exerted better protection than the other two additives, they were selected in subsequent studies for formulation optimization.

### 3.3. Effect of Chelator Combinations on the Color Stability of BRA in Tap Water

To obtain better protection for BRA in tap water, the three selected chelators were combined, and their formulations were optimized through single-factor experimental design. It could be seen from Figure 3 that the three chelators alone could weaken the color decay of the tap water-dissolved BRA solution, and EDTA exhibited the highest anthocyanins retention of 53.69%, followed by SHMP (53.66%) and ST (51.98%) in sequence after storage for 32 h, which was consistent with the visual observation demonstrated in the insets, and the combination of the three chelators further markedly enhanced the protection (Figure 3a–c). It should be noted that the anthocyanins’ retention rates of the BRA solution containing the single chelators kept decreasing with storage proceeding, while that of the BRA solution containing the chelator combinations decreased in the first 8 h or 16 h and then remained nearly constant afterwards. The possible reason might be that the single chelator in the selected concentrations were unable to chelate all the metals, and/or they had higher affinity to other metals than the three selected ones. Hence, Figure 3 demonstrated that the addition of chelator formulations was feasible to further increase the color stability of BRA in tap water. 

The composition of the chelator formulations affected their protection efficiency as well. When EDTA was fixed in the maximum allowed concentration 0.01%, the coexistence of 0.08% SHMP and 0.064% ST conferred the highest anthocyanin retention of 70.54% after storage for 32 h (Figure 3a). Figure 3b revealed that, when SHMP was present in 0.064%, the extra addition of 0.008% EDTA and 0.16% ST contributed the greatest anthocyanin retention of 75.24% after the 32 h storage. Figure 3c revealed that the optimum ST-dominated formulation consisted of 0.008% EDTA, 0.08% SHMP, and 0.096% ST, which conferred an anthocyanin retention of 72.06%. The three optimum formulations, which were designated as F1, F2, and F3, respectively, were investigated in subsequent studies.

### 3.4. Effect of Chelators on the Color of Steamed Cold Noodles

The appearances of steamed cold noodles colored by the three optimized chelator formulations (F1–F3) and the three control steamed cold noodles, including that prepared with tap water and uncolored (designated as NA), colored with deionized water-dissolved BRA (designated as DIA), and colored with tap water-dissolved BRA (designated as TWA), were shown in Figure 4. NA was white in appearance and DIA exhibited the attractive red color. However, TWA turned black, which was consistent with the result in Figure 1a. When the three chelator formulations were added, the resultant noodles exhibited nearly the same color as that of DIA, confirming that the addition of the chelators could increase the color stability of BRA in tap water. 

The color differences of the steamed cold noodles were quantified and the results were shown in Table 1. It could be seen that coloration with tap water-dissolved BRA caused the lowest L*, a*, and b* values among the five colored noodles, confirming that the metals in tap water caused discoloration and decay of BRA, which was consistent with the observation in Figure 4. The addition of the three chelator formulations (F1–F3) restored the chromatic parameters to comparable levels with that of the noodle colored by deionized water-dissolved BRA, among which F1 was the most effective and all the three chromatic parameters were not significantly different from DIA. 

The total color change (ΔE) was calculated by comparing with the steamed cold noodles colored by deionized water-dissolved BRA. It has been reported that a ΔE value less than 3 indicated a color difference unobvious to an untrained observation [30]. As can be seen in Table 1, the ΔE of the noodle colored with tap water-dissolved BRA reached 3.98, indicating that the two noodles differed markedly in color and could be easily distinguished by visual observation. However, the addition of the three chelator formulations reduced the ΔE to below 2, and the lowest value was contributed by chelator formulation F1, implying that nearly no difference in color could be identified by common consumers. Hence, Figure 4 and Table 1 demonstrated that the three chelator formulations were effective BRA protectors regarding the color stability in tap water.

### 3.5. Textural Profile Analysis of Steamed Cold Noodles

Textural properties greatly affect the acceptance of steamed cold noodles by consumers. Table 2 depicted that coloration with BRA had nearly no effect on the texture of steamed cold noodles and nearly no difference could be found between the two colored controls and the uncolored control. However, the addition of the three chelator formulations caused marked textural changes, but no significant difference was found between the three chelator formulations. The greatest changes were observed in the noodle colored with F1, in which, the hardness and gumminess increased markedly to 776.08 g and 186.91 g, respectively, compared with 608.61 g and 125.67 g of the uncolored control. 

In a previous work, it was found that the addition of SHMP increased the hardness and springiness of the noodle made from wheat flour and the authors ascribed this effect to the interaction between SHMP and gluten [31], which was consistent with the current work. Springiness was determined as the degree of recovery of the noodles after the first compression and starch products with higher springiness are often preferred by consumers [32]. Table 2 revealed that the addition of the chelator formulations significantly improved the springiness. Hence, the addition of the chelators to steamed cold noodles could promote its acceptance by consumers especially those that prefer a hard texture.

### 3.6. Retrogradation of Starch in Steamed Cold Noodles

Steamed cold noodles tend to become harder after storage due to starch retrogradation. Hence, the effect of the chelator formulations on this attribute was investigated as well using XRD analysis. As can be seen in Figure 5, all the noodles exhibited the typical peaks of the B-type crystalline structure at 17°–18° and 20° [33], implying that coloration with BRA as well as the presence of the chelators had no effect on the crystal type of the starch. However, the intensities of the two peaks that decreased upon coloration and further declined after the addition of the chelator formulations. The intensity was an indicator of the starch retrogradation process and the higher the intensity, the higher the degree of the process. Hence, Figure 5 clearly revealed that coloration with BRA could effectively slow down the retrogradation of steamed cold noodles and the addition of the chelator formulations could further enhance this effect. This result was consistent with a previous report that the presence of black tea polyphenol extract inhibited the retrogradation of various starches [34] and another report that the proanthocyanidins from grape seeds, peanut skins, and pine barks inhibited the retrogradation of maize starch without changing the crystal type [35].

During retrogradation, amylose molecules became organized in double helices that formed a crystalline network around amylopectin molecules, and the amylopectin molecules became organized in double helices as well [36]. It could be possibly inferred that the interaction between BRA and amylose and/or the side of chain of amylopectin hindered the aggregation of starch molecules by preventing the formation of double-helical structures [35], and the addition of the chelators further weakened the aggregation. Hence, the steamed cold noodles colored with the chelator-containing BRA possessed longer shelf life regarding the starch retrogradation behavior.

### 3.7. In Vitro Digestibility of Steamed Cold Noodles

Figure 6 demonstrated the effect of coloration with BRA and the addition of the chelator formulations on the digestion behavior of steamed cold noodles. The contents of RDS, SDS, and RS in the uncolored noodle was 60.23%, 12.18%, and 27.58%, respectively. Coloration with deionized water-dissolved BRA exerted no influence on the digestive behavior of the noodle, but that, with tap-water, dissolved BRA significantly decreased its RDS content and increased its RS content to 55.17% and 32.74%, respectively, and the addition of the three chelator formulations did not cause further change. No significant difference in digestive behavior was observed between the four steamed noodles made with tap water, implying that the interaction between anthocyanins and starch might be the major reason for the varied digestive behavior. This speculation was consistent with a report that the Mexican blue maize anthocyanins could bind with maize starch through hydrophobic interactions and increase its RS content [37] and another report that coloration with BRA decreased the RDS content of potato starch by restraining its swelling and enhancing its integrity during steaming [38]. All the noodles showed no significance in SDS content. 

The term RS is defined as a certain fraction of starch which arrives in the large intestine almost intact, showing a comparable effect to dietary fiber. Therefore, foods with a high content of RS are regarded as ideal choices for diabetics. The latest research found that the BRA-enriched bread had lower carbohydrate digestibility, reduced glycemic index, and improved postprandial insulin response and could be used as a dietary strategy to control postprandial hyperglycemia [39]. Another work drew a similar conclusion that the interaction with anthocyanins lowed the digestion of rice starch by inhibiting the activity of starch digestive enzymes and the anthocyanins-associated starch could be used to develop foods with low glycemic index [40]. Hence, the steamed cold noodles colored with BRA possibly possess extra health functions.

### 3.8. Antioxidant Capacity of Steamed Cold Noodles

Since anthocyanins possess excellent antioxidant activities, the antioxidant parameters of the steamed cold noodles were measured. As can be seen in Table 3, the uncolored steamed cold noodles did not nearly have any antioxidant activity, but coloration with BRA markedly increased the contents of total polyphenols and anthocyanins as well as the antioxidant capacity, indicating that the interaction with starch could maintain the stability of anthocyanins during cooking. This result was consistent with a report that the anthocyanins-loaded amylopectin nanoparticles assembled in deionized water could protect anthocyanins from heat and oxidization [12].

No significant differences were observed in total polyphenols content between all the colored noodles, but the noodles colored with tap-water dissolved BRA showed significantly lower total anthocyanins contents compared with the deionized water-dissolved control, indicating that the metals in tap water could cause the degradation of anthocyanins. The addition of the three chelator formulations significantly increased the parameter, though they were still lower than that of the control, confirming that the chelators exerted their function by protecting anthocyanins from degradation. Regarding the antioxidant capacity, the uncolored noodles showed only a negligible value due to the presence of trace impurities, while coloration with BRA contributed significant antioxidant capacity increase to the noodles, and a maximum 14.53-fold increase was observed in chelator formulation F1, which did not differ significantly from the deionized water-dissolved control. 

Hence, in addition to imparting an attractive color, BRA also conferred certain antioxidant properties to steamed cold noodles and could be well retained during cooking. Therefore, the colored steamed cold noodles could be a potential carrier for the anthocyanins in BRA, which could be released during colonic fermentation to prevent colonic diseases [41].

## 4. Conclusions

The addition of chelators is a feasible method to increase the color stability of BRA in steamed cold noodles against the Fe^3+^, Fe^2+^, and Cu^2+^ in tap water, and the optimum formulation consisting of 0.01% EDTA, 0.08% SHMP, and 0.064% ST conferred comparable appearance and chromatic attributes to the noodle with that colored by deionized water-dissolved BRA. In addition, coloration with the chelator-supplemented BRA increased the hardness and springiness of steamed cold noodles, delayed its retrogradation, promoted the content of resistant starch, and conferred certain antioxidant capacity. Hence, the chelators-supplemented BRA could be used in the coloration of home-made steamed cold noodles that uses tap water or hard water as the solvent. 

## Figures and Tables

**Figure 1 foods-11-03392-f001:**
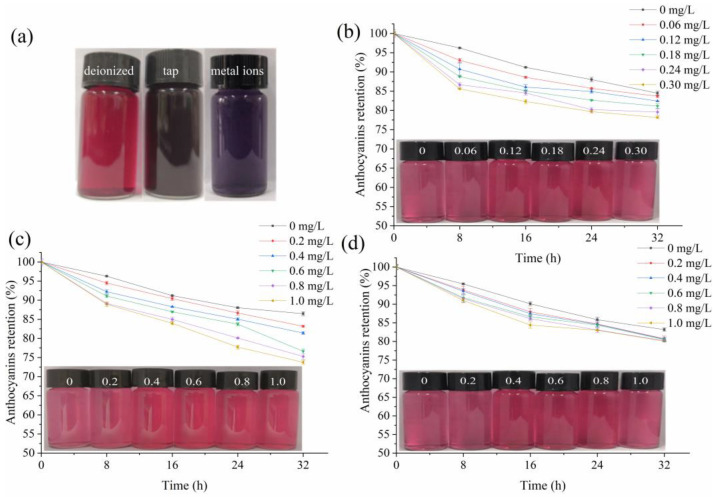
Appearances of the 0.02% (*w/v*) BRA solutions dissolved in deionized water (left), tap water (middle), and deionized water containing the nine metallic ions allowed by GB 5749-2006 (**a**) as well as the color stability of BRA against different concentrations of Fe^3+^ (**b**), Cu^2+^ (**c**), and Fe^2+^ (**d**). The insets are the appearances of the fresh BRA solutions containing the corresponding metals in concentrations from low to high.

**Figure 2 foods-11-03392-f002:**
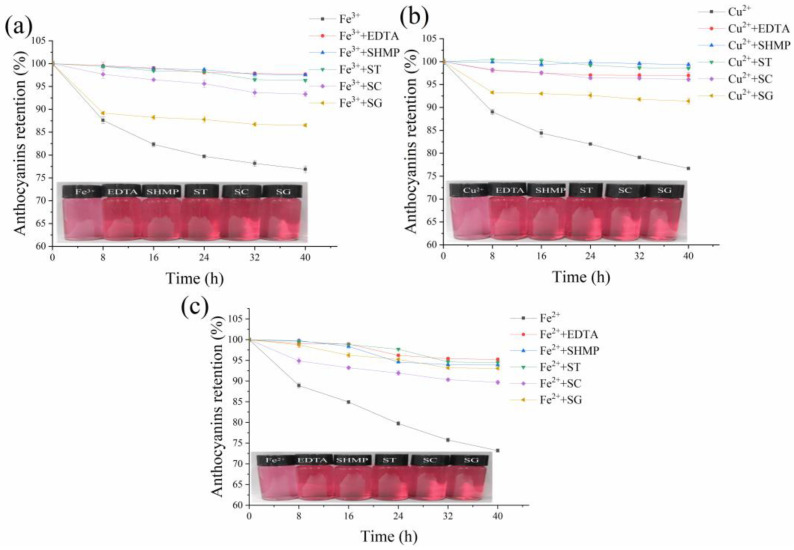
Effects of five chelators, including ethylenediaminetetraacetic acid disodium salt (EDTA), sodium hexametaphosphate (SHMP), sodium tartrate (ST), sodium citrate (SC), and sodium gluconate (SG), on the color stability of the 0.02% (*w/v*) BRA solution against Fe^3+^ (**a**), Cu^2+^ (**b**), and Fe^2+^ (**c**). The insets are the appearances of the fresh BRA solutions containing the corresponding metals and chelators.

**Figure 3 foods-11-03392-f003:**
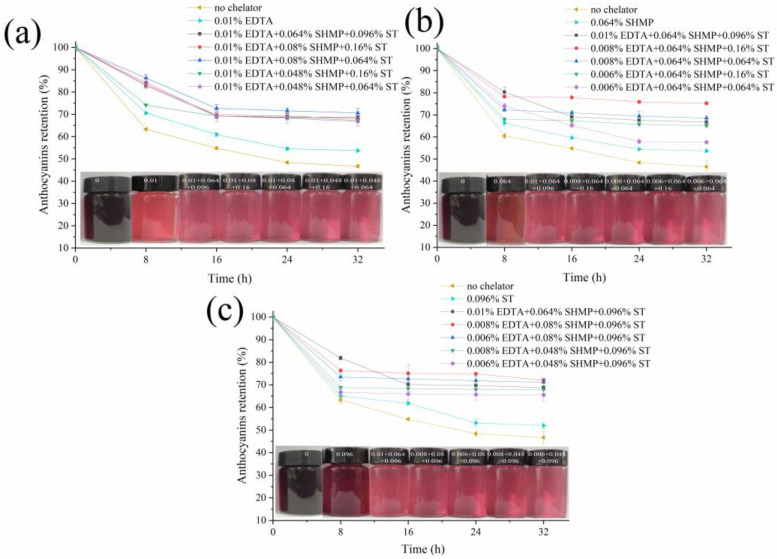
Effect of the combination of ethylenediaminetetraacetic acid disodium salt (EDTA, (**a**)), sodium hexametaphosphate (SHMP, (**b**)), and sodium tartrate (ST, (**c**)) with the other two chelators on the color stability of the tap-water dissolved 0.02% (*w/v*) BRA solution as a function of storage time. The insets are the appearances of the corresponding tap water-dissolved 0.02% BRA solutions containing the chelators formulations after storage for 32 h.

**Figure 4 foods-11-03392-f004:**
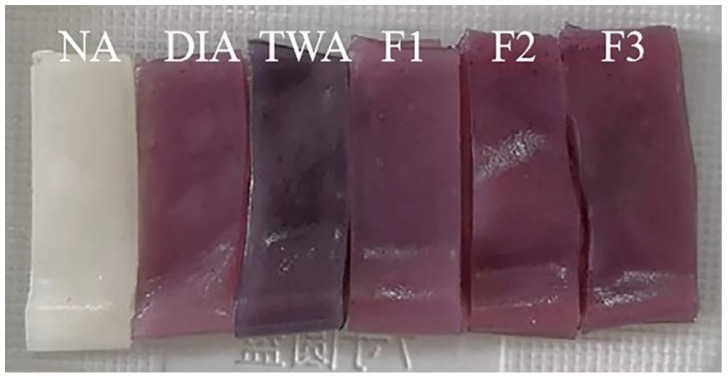
Appearances of the steamed cold noodles that were prepared with tap water and uncolored (NA), colored with deionized water-dissolved BRA (DIA), colored with tap water-dissolved BRA (TWA), and colored with tap water-dissolved BRA containing the F1, F2, and F3 chelator formulations.

**Figure 5 foods-11-03392-f005:**
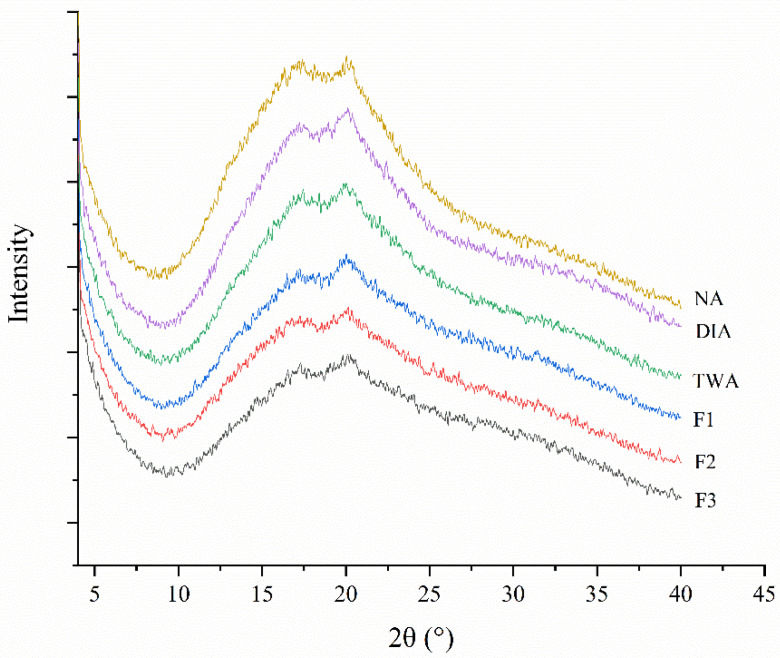
X-ray diffraction spectra of the steamed cold noodles that were not colored (NA), colored with deionized water-dissolved BRA (DIA), colored with tap water-dissolved BRA (TWA), and colored with tap water-dissolved BRA containing the three optimized chelator formulations (F1–F3).

**Figure 6 foods-11-03392-f006:**
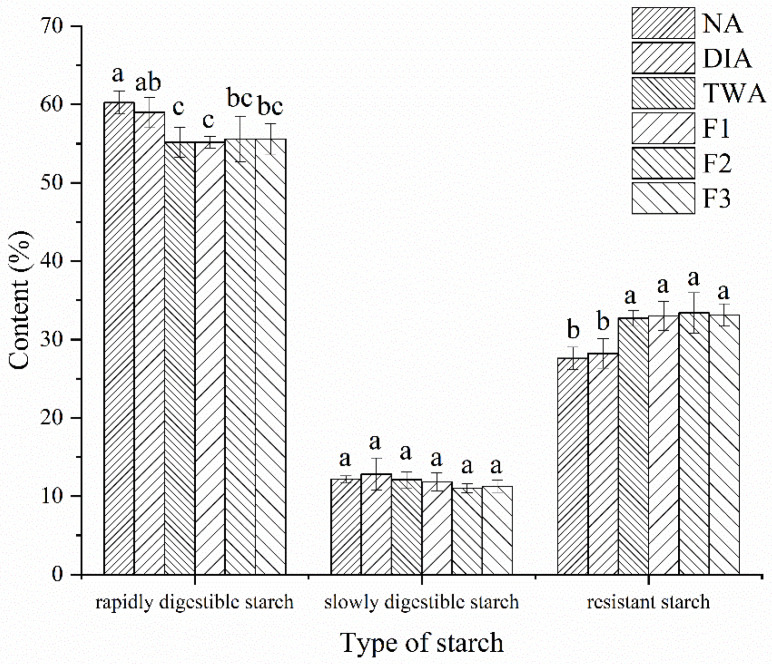
Effect of coloration with BRA and the addition of chelator formulation on the in vitro digestion behavior of the steamed cold noodles that were not colored (NA), colored with deionized water-dissolved BRA (DIA), colored with tap water-dissolved BRA (TWA), and colored with tap water-dissolved BRA containing the three optimized chelator formulations (F1–F3). Values of the same series without the same letter(s) are significantly different at *p* < 0.05.

**Table 1 foods-11-03392-t001:** Chromatic attributes of the steamed cold noodles that were not colored (NA), colored with deionized water-dissolved BRA (DIA), colored with tap water-dissolved BRA (TWA), and colored with tap water-dissolved BRA containing the three optimized chelator formulations (F1–F3).

	L*	a*	b*	ΔE
NA	40.91 ± 3.42 ^a^	0.10 ± 0.02 ^c^	1.21 ± 0.21 ^a^	15.39 ± 0.34
DIA	37.10 ± 2.98 ^ab^	14.74 ± 0.38 ^a^	−1.53 ± 0.43 ^c^	-
TWA	33.70 ± 3.00 ^b^	12.70 ± 0.42 ^b^	−1.78 ± 0.23 ^c^	3.98 ± 0.03
F1	36.69 ± 3.10 ^ab^	14.59 ± 0.94 ^a^	−1.61 ± 0.45 ^c^	0.61 ± 0.24
F2	36.52 ± 2.38 ^ab^	14.46 ± 0.32 ^a^	−0.61 ± 0.19 ^b^	1.24 ± 0.15
F3	35.95 ± 2.83 ^ab^	13.15 ± 0.73 ^b^	−1.45 ± 0.24 ^c^	1.99 ± 0.20

Note: Values of the same column without same letter(s) are significantly different at *p* < 0.05.

**Table 2 foods-11-03392-t002:** Textural profiles of the steamed cold noodles that were not colored (NA), colored with deionized water-dissolved BRA (DIA), colored with tap water-dissolved BRA (TWA), and colored with tap water-dissolved BRA containing the three optimized chelator formulations (F1–F3).

	Hardness (g)	Springiness	Gumminess (g)	Cohesivenes	Chewiness	Resilience
NA	608.61 ± 16.70 ^b^	0.19 ± 0.01 ^a^	125.67 ± 4.12 ^d^	0.22 ± 0.01 ^bc^	23.87 ± 3.02 ^bc^	0.03 ± 0.00 ^b^
DIA	601.86 ± 9.98 ^b^	0.19 ± 0.00 ^a^	139.53 ± 4.45 ^c^	0.23 ± 0.00 ^b^	26.66 ± 1.59 ^ab^	0.04 ± 0.00 ^ab^
TWA	609.45 ± 51.97 ^b^	0.19 ± 0.01 ^a^	126.98 ± 6.67 ^d^	0.21 ± 0.01 ^c^	23.34 ± 2.51 ^c^	0.03 ± 0.00 ^b^
F1	776.08 ± 46.81 ^a^	0.21 ± 0.00 ^b^	186.91 ± 4.72 ^a^	0.25 ± 0.01 ^a^	28.28 ± 1.00 ^a^	0.04 ± 0.00 ^ab^
F2	755.74 ± 24.53 ^a^	0.21 ± 0.00 ^b^	181.23 ± 3.75 ^ab^	0.25 ± 0.00 ^a^	26.83 ± 0.35 ^ab^	0.04 ± 0.00 ^ab^
F3	756.43 ± 60.18 ^a^	0.22 ± 0.00 ^b^	175.92 ± 3.46 ^b^	0.25 ± 0.00 ^a^	27.92 ± 1.11 ^a^	0.05 ± 0.00 ^a^

Note: Values of the same column without same letter(s) are significantly different at *p* < 0.05.

**Table 3 foods-11-03392-t003:** Total polyphenols content, total anthocyanins content, and antioxidant capacity of the steamed cold noodles that were not colored (NA), colored with deionized water-dissolved BRA (DIA), colored with tap water-dissolved BRA (TWA), and colored with tap water-dissolved BRA containing the three optimized chelator formulations (F1–F3).

	NA	DIA	TWA	F1	F2	F3
Total polyphenols (mg·(g)^−1^)	0.00 ± 0.00 ^b^	0.03 ± 0.01 ^a^	0.03 ± 0.00 ^a^	0.03 ± 0.01 ^a^	0.03 ± 0.00 ^a^	0.03 ± 0.00 ^a^
Total anthocyanins (mg·(kg)^−1^)	0.00 ± 0.00 ^e^	24.19 ± 0.14 ^a^	20.85 ± 0.09 ^d^	23.78 ± 0.09 ^b^	23.37 ± 0.05 ^c^	23.45 ± 0.37 ^c^
Antioxidant capacity (%)	1.37 ± 0.54 ^c^	20.21 ± 0.14 ^a^	19.49 ± 0.43 ^a^	19.91 ± 0.77 ^a^	14.70 ± 0.31 ^b^	15.01 ± 0.52 ^b^

Note: Values of the same column without same letter(s) are significantly different at *p* < 0.05.

## Data Availability

Not applicable.

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
