# Peer review of "Addition of Chelators Increased the Stability of Black Rice Anthocyanins against the Metallic Ions in Tap Water and Improved the Coloration of Steamed Cold Noodles"

_foods, 2022, doi:10.3390/foods11213392_

Round 1

Reviewer 1 Report

Zheng et al. have demonstrated in their study that addition of chelators could increase the stability of black rice anthocyanins against metallic ions in steamed cold noodle preparation. It is good work that is well planned, executed and presented in a lucid way. The article contains all the scientific elements to demonstrate the hypothesis proposed in the objective. Therefore, I recommend responding to the following minor comments:

1.       Keywords – one more keyword “antioxidant activity” should be added.

2.       Section 2.8 – a reference should be included for the texture analysis method.

3.       Section 2.13 – there is no need of “%” symbol on the right-hand side of the DPPH formula.

4.       Section 2.14 – the statistical software SPSS should accompany state, city and country name in the case of USA.

5.       Lines 246-249 – a reference for “allowed chelators” by China should be included, which can be either a government document release or publication etc.

Reviewer 2 Report

·        Line 27. Please correct the word “attract-ing”

·        Line 112. Please correct the word “solu-tion”

·        Line 229. Please use superscript in “Cu2+”

·    In Figure 1. It says “The  insets are the appearances of the BRA solutions containing the corresponding metals in concentrations from low to high.” Please indicate the storage time belonging to the pictures.

·        In the “Standards for Drinking Water Quality”, Fe is allowed maximum 0.3mg/L, Cu is allowed maximum 1mg/mL. In Figure 1b, c, and d, the concentrations of metal ions in the solutions are not comparable with each other. In 1b, the Fe3+ concentrations were between 0.06-0.30 mg/L but in 1c, and 1d, the Fe2+ and Cu2+ concentrations were between 0.02-1mg/L. Please explain the reasons of differences.

·        Line 237. The color change of the extract containing 9 metal ions in Figure 1a was attributed to “synergistic fading effect” of the metal ions on color. Unfortunately, any experiment was conducted to determine this effect. It would be good to rearrange the statement in the sentence to include probability of additive effect or synergistic effect.

·        Line 241-245. This section could be moved into an “Introduction” part.

·        Line 283. Please correct the unit of storage time “for 32 d”.

·        In Figure 2, please indicate the storage time of the extracts whose pictures given in the figures.

·        In Table 1. Please remove “Values of the same column without same letter(s) are significantly different at p < 0.05” from the headline. You have already given the same explanation as “Note” below the table. The similar problem also existed in Table 2.

·        Line 364-365. Please give a reference for this explanation.

·        The sensorial tests are missing. Please give some information why sensorial tests were not done by panellists.

Reviewer 3 Report

The manuscript is interesting, describing a process to increase the stability of anthocyanins added in steamed cold noodle preparation by the addition of chelators agents. For a better understanding, some points must be taken into consideration before publication.

 Line 177: …gallic acid equivalents (CE) should be “gallic acid equivalents (GAE)”.

Results:

Please improve the resolution of Figs. 1-3 and add the error bars for each treatment. You could add a table to summarize the results of figs 1-3.

 Please improve the resolution of Figs. 1-3. Also, consider adding a table to summarize the results of Figs 1-3 with the mean comparison test, in order to clarify the discussion of results and how the best treatments were selected for the following experiments.

 Line 224: Consider that the treatments lose color due to other factors such as exposure to light, pH and temperature. In the control (Fig. 1b), a retention of 85% is observed, while, in the presence of Fe+3 at different concentrations, the values are from 80 to 84%, it is recommended to present the change due to the effect of the addition of the metal ion. The effect of the metal ion is greater when Cu is used.

 Line 232: What were the criteria for selecting these concentrations? Is there a significant difference between treatments?

 Table 3: To what do you attribute the antioxidant activity in the control?

 On what basis to conclude that 20% of antioxidant activity is considered to be excellent? There is some report to compare? I think that the conclusion should focus on the fact that F1 increases up to 14.53-fold the antioxidant activity of noodles, with no changes in appearance. 
